# Towards Modelica Models with Credibility Information

**Martin Otter** [1],*[ID]**, Matthias Reiner** [1]**, Jakub Tobolář** [1][ID]**, Leo Gall** [2] **and Matthias Schäfer** [2]

1 German Aerospace Center (DLR), Institute of System Dynamics and Control (SR), 82234 Wessling, Germany
2 LTX Simulation GmbH, 80939 Munich, Germany
* Correspondence: martin.otter@dlr.de

**Abstract:** Modeling and simulation is increasingly used in the design process for a wide span of applications. Rising demands and the complexity of modern products also increase the need for models and tools capable to cover areas such as virtual testing, design-space exploration or digital twins, and to provide measures of the quality of the models and the achieved results. The latter is also called credible simulation process. In an article at the International Modelica Conference 2021, we summarized the state of the art and best practice from the viewpoint of a Modelica language user, based on the experience gained in projects in which Modelica models were utilized in the design process. Furthermore, missing features and gaps in the used processes were identified. In this article, new proposals are presented to improve the quality of Modelica models, in particular by adding traceability, uncertainty, and calibration information of the parameters in a standardized way to Modelica models. Furthermore, the new open-source Modelica library Credibility is discussed together with examples to support the implementation of credible Modelica models.

**Keywords:** credible model; model requirement; data management; validation; verification; Modelica model

## 1. Introduction

The modeling of physical systems is a complex process with many decisions, simplifications and assumptions along the way, usually done by many different decision-makers. In the real world, this often leads to simulation models and simulation results that are not well documented. Due to the increasing use of system simulation in today's product development, there is an increasing need for traceable model development with guarantees on model validity.

Models are often shared across organizational borders (for example, from supplier to OEM). On the way, the direct access to model sources (for example to internal repositories) and the model history is lost. To avoid this, models have to "carry" documentation "with them". If only black-box Functional Mock-up Units (FMUs) (https://fmi-standard.org (accessed on 30 July 2022)) [1–3] or result data with simulation reports are shared, the requirements on credibility and traceability are even higher.

The Modelica language (https://modelica.org/modelicalanguage.html (accessed on 30 July 2022)) [4,5] is a standardized, nonproprietary, equation-based, object-oriented language to model multidomain systems. Mature open-source and commercial tools support Modelica (https://modelica.org/tools.html (accessed on 30 July 2022)) and a large number of open-source and commercial Modelica libraries are available (https://modelica.org/libraries.html (accessed on 30 July 2022)). For a comparison of Modelica with other multidomain modeling languages, see [6].

The ITEA 3 project UPSIM (https://itea3.org/project/upsim.html and https://www.upsim-project.eu (accessed on 30 July 2022)) aims for system simulation credibility via introducing a formal simulation quality management approach, encompassing collaboration and continuous integration for complex systems. It shall be based on the recently proposed *"Credible Simulation Process"* [7,8] of the SetLevel project (https://setlevel.de/en (accessed

on 30 July 2022)). In [9], the state of the art in the development and management of credible Modelica models was summarized and missing features in the used processes were identified to guide further research in the UPSIM project. Based on this analysis, new proposals are presented below to improve the quality of Modelica models, in particular by adding traceability, uncertainty and calibration information of the parameters in a standardized way to models. The suggestions are provided first in a neutral way that should be valid for any kind of steady-state or dynamic models. Afterwards, it is shown how to concretely implement Modelica models with credibility information by utilizing the new open-source Modelica library *Credibility*. Several examples demonstrate this approach.

There is a well-established foundation for several aspects of credible models, see for example [10]. Furthermore, there is the huge area of verification and validation of models, see for example the extensive literature overview in [11]. Many simulation environments provide methods for calibration, verification, validation, uncertainty analysis, sensitivity analysis, etc. The usual approach is to start with a parameterized model and add model-specific information for the analysis to be carried out. For example, statistical properties of parameters might be added to a Modelica model as proposed in [12] and this information might be used to calibrate a model using data reconciliation [12,13]. The focus of this article is not on new methods in these areas but on how to associate information about the credibility of a Modelica model in a standardized, practical way to a model definition, so that it is inherently available when the model is archived or distributed.

There are existing and/or upcoming standards in various industries that have similar goals but with a partially different focus:

- In [14], it was shown how the traceability information of Modelica models was exchanged by OpenModelica with other lifecycle tools through the standardized *Open Services for Lifecycle Collaboration* (OSLC) (https://open-services.net/ (accessed on 30 July 2022)) interface.
- The NASA *Handbook for Models and Simulations* [15] consists of a report (NASA-STD-7009A W/CHANGE 1) containing a high-level description of requirements, recommendations and best practices for the development of credible models and their usage, as well as two Excel files—*M&S Life-Cycle Worksheet*, *Requirements and Recommendations per M&S Life Cycle Phase*—to practically apply them.
- The prime objective of the LOTAR (LOng Term Archiving and Retrieval) International Consortium [16] "*is the creation and deployment of the EN/NAS 9300 series of standards for long-term archiving and retrieval of digital data, based on standardized approaches and solutions*" in the aerospace and defense industry. The LOTAR MBSE workgroup suggests the usage of the Modelica, FMI (https://fmi-standard.org (accessed on 30 July 2022)) and SSP (https://ssp-standard.org (accessed on 30 July 2022)) standards of the Modelica Association (https://modelica.org/ (accessed on 30 July 2022)) as a basis and intends to extends them with LOTAR manifest files [17] that hold, for example, information such as model usage, validity ranges, represented and neglected phenomena.

## 2. Model Credibility

A large part of this paper focuses on the credibility of parameter values, because the added information can directly be used for simulation tasks. Therefore, we need first to discuss the documentation of system models and subsystems which use such credible parameter values, even if this is a less technical topic. The Modelica language already provides specific annotations for documentation, revisions and version. In Section 2.1, we discuss what kind of information is needed in the model documentation and in Section 2.2 how it can be stored in user-defined metadata attached to the model.

### 2.1. Model and Process Quality

In order to assess model credibility, we start with the following thought experiment: You receive a model from someone to whom you cannot ask questions, because he/she left the company. Do you trust this model? How much effort is it for you to perform further

simulation studies based on this model? Do you understand the modeling assumptions used when the model was created? This trust into the results is built on the answers to different questions:

- Who developed the model in the first place?
- For which task was it developed?
- Was it useful in other projects?
- How has it been validated?
- Do we just know that the model is useful or is there written proof via documentation, validation reports, etc.?

These questions can be grouped into two different categories:

On one hand, there is the *artefact quality*, meaning a documented state of the model code with all its resources (measurements, boundaries, etc.) is available. The second category is the *process quality*, meaning a documentation of the development steps of the model, the steps to reproduce specific simulation results and the quality of these results.

There are various ways to define the quality of the simulation results mostly based on a verification and validation process based on reference data of models, as, for example, described in [18,19], but to our knowledge there is no commonly accepted way to define it.

We have to keep in mind that a model is always developed for a specific simulation task. For example, a model of a cooling circuit could be developed for determining different temperatures inside the circuit. No detailed modeling of hydraulic effects is necessary in such a case. However, this model is not suited to determine pressures inside this cooling circuit (which is another simulation task). So, the credibility has to be reassessed for each simulation task, in order to trust the results of the simulation for the current scenario.

*2.2. Model Metadata*

The information about artefact quality, as well as process quality, needs to be accessible for anyone working with the model. In order to guarantee this access, we propose to have this documentation stored as metadata inside the model itself. This should avoid trouble with storing and linking different files and offer the possibility to apply automatic methods to evaluate a model quality measure based on this documentation. Storing metadata within the model is conceptually similar to the model documentation that is stored in annotations within the Modelica model file.

The artefact and process quality and further information can be stored in the metadata in a structured way. For this, we identified three main categories of information necessary for a new developer to trust a model and to work with it. The categories are as follows:

- The model itself (modeling aim, modeling assumptions, range of validity, etc.).
- The simulations performed (hardware information, scripts for execution, accessed external data, etc.).
- The tool used (tool version, settings, etc.).

Each of these main categories can be further structured by subcategories, till a hierarchical level is reached, where each information can be stored as key–value pair, similar to the Simulation Resource Meta Data (SRMD) format proposed in [7]. This structure also allows references to external data such as measurements or supplier data sheets. A first proposal on how to store a minimal set of metadata is shown in Figure 1. For the corresponding Modelica code, see Listing A1 in Appendix A.

In order to motivate model developers to create these metadata and keep them up-to-date, they must be reachable, easy to understand and requiring a small extra effort. To reduce the effort, most of the entries listed in Figure 1 could be filled automatically by the simulation tool or by a user- or project-specific reference configuration. Therefore, storing the information inside the model seems feasible, because no external tools or files are needed. In a Modelica model, these metadata are usually stored in annotations. When exporting a Modelica model in FMI format, annotations are typically stored in the FMI

modelDescription.xml file and therefore the metadata information can be accessed by FMI tools.

| Key | Description |
|---|---|
| Author* | John Doe |
| Organization* | My Company |
| Contact | john.doe@mycompany.domain |
| CreationDate* | 2022-07-04 |
| ModifiedDate | 2022-07-29 |
| CreatedInProject | UPSIM 2022 |
| License | BSD 3-clause |
| MSLVersion | 4.0.0 |
| CopiedFromModel | Modelica.Electrical.Machines.Examples.Transformers.Rectifier6pulse |
| CopiedFromModelVersion | 4.0.0 |
| CredibilityLevel | 0 |
| Checked | false |
| CheckedBy | |
| ModelicaTool | Dymola |
| ModelicaToolVersion | 2023 |
| ModelicaLanguageVersion | 3.5 |

**Figure 1.** Screenshot of user-defined metadata in Dymola, with a proposed set of keys (* means a required key; otherwise, the key is optional).

We identified three challenges in the handling of metadata. (i) There need to be conventions, or clearly defined rules, when an entry needs to be updated. This can be illustrated by the simple question of who the author of a model is: the developer of the first draft of the model, the one who has performed the last update or anyone in between? (ii) Furthermore, it needs to be defined whether metadata can be inherited from other models, e.g., from components used in the model. (iii) The mandatory entries of metadata must be defined depending on the required credibility level, which can vary between different components of a hierarchical model. When exporting a credible model from the simulation tool in FMI or SSP format, less detailed metadata might be sufficient to be exported for lower hierarchical levels than for the top-level model. As a conclusion, we can state that storing a specific set of metadata in the source model (e.g., directly in Modelica) seems promising. However, more work is needed for integrating it nicely into simulation processes and corresponding tool chains, especially dedicated product data management and traceability tools.

### 3. Parameter Credibility

For a credible model it is not sufficient to only store information about the whole model in the metadata as described in Section 2, but a documentation of each parameter should also be available, including credibility information. In the following, we introduce such type of credibility information for parameters. According to this approach, each parameter of a Modelica model does not only define a scalar or an array value, but the parameter value and additional information concerning *traceability, uncertainty* and *calibration* are stored in a *record* data structure. The three categories are described in Sections 3.1–3.3, respectively.

An example of the type of information stored in a parameter record is shown in Figure 2. The parameter record can be extended with more information as needed for the parameter.

```
c = 1400.8
└
  ┌ min    = 0
  │ max    = 1e6
  │ unit   = "N.m/rad"
  │ description = "Compliance spring constant"
  │
  │
  │  Traceability
  │    source   = Calibrated
  └    info     = "Based on measurements on test bed B in project XYZ."
       reference = "modelica://ProjectXYZData/ReportXYZ_B17.pdf"
     Uncertainty
       source      = Estimated
       type        = TruncatedNormalTolerance
       nominal     = 1400.8
       relTol      = 0.05    // 5% relative tolerance
       stdDevFactor = 3       // 99.7% at the limits of underlying normal distribution
     Calibration
       start       = 1000
       min         = 100.0
       max         = 5000.0
       __ToolA_Setup = "modelica://ProjectXYZLib/Resources/calibrateDrive.mos"
```

**Figure 2.** Example of the structure of the parameter record for a scalar parameter c.

*3.1. Traceability*

The first category is the traceability of a parameter and its value. Here, information shall be provided on why a parameter has a specific value. This is defined with the three keys:

- *source*—origin of the parameter value (e.g., *Estimated*), see enumeration below.
- *info*—a short textual explanation how the value was determined (optional).
- *reference*—identifies the resource where the determination of the value is described, for example, a web address, DOI, ISBN, internal report ID of a document, data sheet, etc., (optional).

Hereby, *Source* represents the way the value of the parameter was determined. It is an enumeration that can take one of the following values

- *Unknown*—the source of the value is not known.
- *Estimated*—value from an educated guess, previous projects, extrapolated value, etc.
- *Provided*—value from a data sheet, supplier specification, etc.
- *Computed*—value from computer-aided data such as CAD, CFD, FEM, etc.
- *Measured*—measured value from physical measurements.
- *Calibrated*—value from a model calibration (based on provided data).

Obviously, parameters without a given source (source = *Unknown*) have to be handled with care and are not sufficient for the demand of a good traceability. In case the parameter value is based on a recorded source (such as *Provided*, *Computed* or *Measured*) there need to be a *reference* to this source (e.g., the data sheet location in the local file system or on the web).

The referenced resource must be stored in a way that avoids accidental manipulation, e.g., in a version control system with a given hash, in order to retain the reference information reliably. In the case of calibrated data, the setup of the calibration—this means all information needed to repeat exactly this calibration—and the allowed parameter range should be stored in the calibration section of the parameter record, see Section 3.3 for more details.

*3.2. Uncertainty*

An essential part of a credible model is the information about the quality of a parameter value. This is typically expressed as the *uncertainty* of the parameter. Besides a mathematical description of the uncertainty (see below), the same information as for the traceability of its value also needs to be given (see Section 3.1):

- *source*—origin of the uncertainty determination (e.g., *Estimated*), see enumeration in Section 3.1.
- *info*—a short textual explanation on how the uncertainty was determined (optional).
- *reference*—identifies the resource where the uncertainty determination is described, for example, a web address, DOI, ISBN, internal report ID of a document, data sheet, etc., (optional).

The goal in the remaining part of this section is the mathematical description of uncertainties of scalars in Section 3.2.1, and of arrays/tables in Section 3.2.2. There is a huge literature on the *mathematical description* of uncertainties, see for example [11,20–23]. The article of Riedmaier et al. [11] provides a comprehensive literature overview. In [12], some language constructs were proposed to describe uncertain values in the Modelica language. For linear models, structured and unstructured uncertainties, including uncertainties of unmodeled dynamics, were described in various ways, such as LFT (linear fractional transformation), see for example [24]. These description forms are not used here, because this section discusses methods to model nonlinear systems with parametric uncertainties.

The treatment below is focused on providing the information in a way that it can be used by models described with the Modelica language [4], FMI [1–3], SSP [25], Modia [26] or similar modeling approaches. Furthermore, not only physical/measured quantities are considered, but the same description form is also used to define *requirements* or *specifications*. For example, a simulation result must match a *reference result with some uncertainty*, or the result of a design must match some *criteria* that are specified with an *uncertainty* description.

3.2.1. Uncertainties of Scalars

Every physical quantity has inherent limits. Therefore, the upper and lower limits of a scalar value need to be defined, independent of the kind of the mathematical description of the uncertainty. Often, for example, an uncertainty of a physical quantity is described by a *normal distribution*. However, this does not make sense because this distribution defines probabilities for values in the range from $-\infty$ to $+\infty$. Assume for example, a resistance of an electrical resistor is described with such a normal distribution and a Monte Carlo simulator selects a random value according to this distribution that is negative (although the probability might be very small), then the corresponding simulation makes no sense at all, because the physical resistance can only have positive values.

Furthermore, if a real-world product is used and certain parameters of this product are defined with tolerances or belong to certain tolerance classes according to some standard (for example a resistor with a tolerance of $\pm 5$ % for the resistance value), then the manufacturer guarantees that the parameter value is within the specified tolerance. For this reason, the minimum information to be provided for a scalar parameter with an uncertainty description of any kind is

- *nominal*—the nominal value of the scalar (e.g., determined by calibration).
- *lower*—the lowest possible value of the uncertain scalar.
- *upper*—the highest possible value of the uncertain scalar.

For example, the parameter $R$ of a resistor has $nominal = 200\,\Omega$, $lower = 190\,\Omega$ and $upper = 210\,\Omega$. It is also necessary to define the *value* of the scalar in a model that is used in the current simulation run, for example, the value used in an optimization run that determines improved control parameters, or the value used during a Monte Carlo simulation to propagate uncertainties to output variables. Typically, the *nominal* value is used as default for the current *value*.

Often, it is inconvenient to provide absolute ranges and instead, relative or absolute deviations are more practical. In the eFMI (Functional Mock-up Interface for embedded systems) standard (https://emphysis.github.io/pages/downloads/efmi_specification_1.0.0-alpha.4.html (accessed on 30 July 2022)) (Section 3.2.4) *tolerances* for *reference results* are defined in a similar way as tolerances for numerical integration algorithms. Due to its generality, this description form of eFMI is used here as well:

- *nominal*—nominal value of the scalar.
- *relTol*—relative tolerance of limits with respect to *nominal* (default = 0.0).
- *absTol*—absolute tolerance of limits with respect to *nominal* (default = 0.0).

The *lower* and *upper* values can be computed from these parameters in the following way:

$$tol = \max(absTol, relTol \cdot |nominal|) \tag{1}$$
$$lower = nominal - tol \tag{2}$$
$$upper = nominal + tol \tag{3}$$

Examples:

- $200\,\Omega \pm 5\%$—limits defined with $relTol = 0.05$ ($absTol = 0$).
- $200\,\Omega \pm 10\,\Omega$—limits defined with $absTol = 10\,\Omega$ ($relTol = 0$).

Typically, either a description with *relTol* or with *absTol* is used, but not a combination of both. The alternative is therefore to introduce a flag to distinguish whether *relTol* or *absTol* is provided. However, the drawback is that an additional flag needs to be introduced and that one of the values is always ignored (and any given value might be confusing). Furthermore, there is an important use case, where the description with both *relTol* and *absTol* is useful: If *reference results* are provided (e.g., the results of a simulation or an experiment are required to be within some band around a provided reference solution), then a definition with *relTol* is often practical together with *absTol* as a band for small values of the reference around zero, where *relTol* makes no sense.

The minimum information for an uncertainty description of a model parameter was defined above. Further properties can be defined due to numerous reasons why a model parameter is uncertain. The following list sketches some reasons for uncertainties according to [20]:

- Systematic errors, calibration errors, etc., in the measurement equipment;
- Uncontrolled conditions of the measurement equipment (e.g., environment);
- Differences in the parameter values of the same device, e.g., due to small variations in the production or the used materials;
- Mathematical model does not reflect the physical system precisely enough;
- Not enough knowledge about the physical system;
- Uncontrolled conditions for the model inputs/environment and scenarios.

Uncertainties are typically classified as either *aleatory* or *epistemic*. [20] provides the following definitions:

- *Aleatory — the inherent variation in a quantity that, given sufficient samples of the stochastic process, can be characterized via a probability density distribution.*
- *Epistemic—uncertainty due to lack of knowledge by the modelers, analysts conducting the analysis, or experimentalists involved in validation.*

An uncertainty can also consist of a combination of (a) and (b). An epistemic uncertainty can be reduced through increased understanding, or increased and more relevant data, whereas the statistical property of an aleatory uncertainty is usually inherent in the system and can thus not be reduced (this would require, e.g., to improve a production process).

An epistemic uncertainty is defined by an *interval*. Besides the classification as interval, the above minimum description of an uncertainty by *lower*, *upper* (besides *nominal*) values is sufficient to define this type of uncertainty. Hereby, no probability distribution is known due to a lack of knowledge. Typical epistemic uncertainties are the load of a vehicle, aircraft or robot. Interval arithmetic or Monte Carlo samples might be used for analysis, see [27]. In the latter case, approximated intervals are computed for interested output variables, but no stochastic distributions are determined as for aleatory uncertainties. If both epistemic and aleatory uncertainties are present, an interval might be divided in a grid and for every grid value, a Monte Carlo simulation is performed for the aleatory uncertainties. An interested

output variable is computed as a *set of probability distributions* and presented as a probability distribution area, see [22].

Aleatory uncertainties with limits are defined by *truncated probability distributions* (https://en.wikipedia.org/wiki/Truncated_distribution (accessed on 30 July 2022)) [28]. A distribution of this kind is derived from an underlying (standard/nontruncated) probability distribution. The Modelica Standard Library (https://github.com/modelica/ModelicaSt andardLibrary (accessed on 30 July 2022)) provides *TruncatedNormal* and *TruncatedWeibull* distributions (https://doc.modelica.org/Modelica4.0.0/Resources/helpDymola/Modeli ca_Math_Distributions.html (accessed on 30 July 2022)). About 60 univariate distributions together with their truncated versions are available from Julia package *Distributions.jl* (https: //juliastats.org/Distributions.jl/stable (accessed on 30 July 2022)). It is straightforward to map these definitions to any other modeling or programming language. In this article, we propose to define an uncertainty of this kind by the minimum description of an uncertainty as above, together with the type of the distribution and additional distribution specific parameters, see Table 1.

**Table 1.** Examples for the proposal to define uncertainty descriptions with generic and tolerance parameterizations of scalar parameters. Default values are given with "=…", such as "*relTol* = 0". A truncated normal distribution is defined by the *nominal* value (= mean value) and the standard deviation *stdDev* of the underlying normal distribution, besides *lower* and *upper* limits.

| Uncertainty Name | Parameters | Uncertainty Type |
|---|---|---|
| *Interval* | *nominal, lower, upper* | Epistemic |
| *Uniform* | *nominal, lower, upper* | Uniform distribution |
| *TruncatedNormal* | *nominal, lower, upper, stdDev* | Truncated normal distribution |
| *IntervalTolerance* | *nominal, relTol* = 0, *absTol* = 0 | Epistemic |
| *UniformTolerance* | *nominal, relTol* = 0, *absTol* = 0 | Uniform distribution |
| *TruncatedNormalTolerance* | *nominal, relTol* = 0, *absTol* = 0, *stdDevFactor* = 3 | Truncated normal distribution |

The parameters of the *TruncatedNormal* distribution are computed from the proposed *TruncatedNormalTolerance* parameterization in the following way:

$$tol = \max(absTol, relTol \cdot |nominal|) \tag{4}$$

$$lower = nominal - tol \tag{5}$$

$$upper = nominal + tol \tag{6}$$

$$stdDev = tol / stdDevFactor \tag{7}$$

Examples of the uncertainty description of a resistance *R* with this parameterization are given in Table 2. Specifications are sometimes given in the form of the first column in this table, which was the motivation for the specific (intuitive) parameterization via *TruncatedNormalTolerance*. The corresponding probability density functions are shown in Figure 3.

**Table 2.** Examples of *TruncatedNormalTolerance* definitions. The value $2\sigma$ means that the limits of the truncated normal distribution are at $2 \cdot standardDeviation$ (= probability of 95.4%) of the underlying (nontruncated) normal distribution, see also Figure 3.

| Specification | Parameters of TruncatedNormalTolerance |
|---|---|
| $200\,\Omega \pm 5\%$, $2\sigma$ | *nominal* = 200, *relTol* = 0.05, *stdDevFactor* = 2 |
| $200\,\Omega \pm 10\,\Omega$, $2\sigma$ | *nominal* = 200, *absTol* = 10, *stdDevFactor* = 2 |

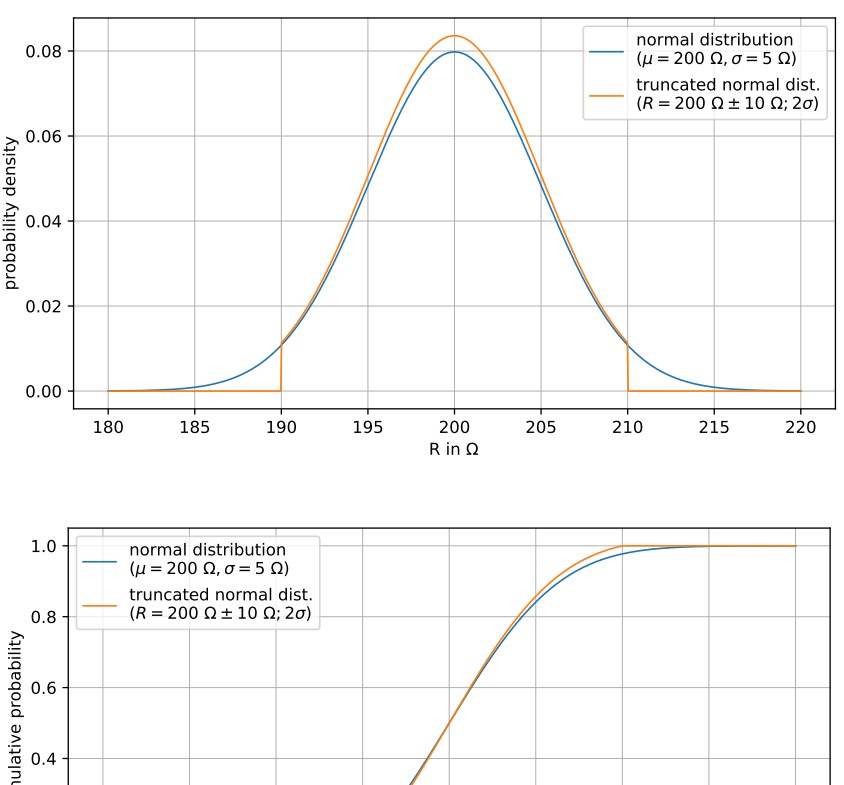

**Figure 3.** Probability density and cumulative probability of the *truncated normal distribution* (orange curve) of resistance $R = 200\,\Omega \pm 10\,\Omega$, $2\sigma$ such that the underlying (nontruncated) *normal distribution* (blue curve) has mean value $\mu = 200\,\Omega$ and standard deviation $\sigma = 5\,\Omega$ ($= 10\,\Omega/2$). The probability is 100% for the truncated normal distribution that a value of $R$ is between $190\,\Omega$ and $210\,\Omega$ and the probability is 95.4% for the underlying normal distribution for this range.

There are various standards of technical systems to describe uncertainties of a product:

- Tolerances of resistors and capacitors are defined by IEC 60062:2016 (https://en.wikipedia.org/wiki/RKM_code (accessed on 30 July 2022)).
- Tolerances of geometrical products are defined by ISO 286-1:2010 (https://www.iso.org/standard/45975.html and https://en.wikipedia.org/wiki/Engineering_tolerance (accessed on 30 July 2022)).

If an industrial standard is used that only defines tolerances (without a statistical distribution), the corresponding uncertainty would be typically defined as an *IntervalTolerance* or *Interval* uncertainty.

The determination of the uncertainty distribution of a parameter might be difficult, if the parameter value cannot be directly measured. In [29], new methods were proposed and evaluated, e.g., to infer the uncertainty distributions of model parameters of a DC-motor from test-bench measurements.

### 3.2.2. Uncertainties of Arrays

Submodels are often approximated by *tables of a characteristic property* that have been determined by measurements, for example a table defining friction torque as a function

of the relative angular velocity, or a table defining mass flow rate through a valve as a function of the valve position. Tables of this kind are basically defined with two or more dimensional arrays. *Inputs of a system* are often defined by tables as a function of time. The *outputs of a system* might be checked against reference outputs (for example, determined from the previous version of the model or tool, from another tool or from an analytically known solution) defined by tables as a function of time. The computed solution is then required to be within the uncertainty ranges of the reference tables. All these examples have parameter arrays, where the uncertainties of the array elements need to be described.

If a characteristic is determined by detailed measurements, then typically, for every element, the upper and lower limits are known from the available measurement data, see for example [30] for a force–velocity dependency of automotive shock absorbers. This means that the measurements can be summarized by a *nominal array* that has been determined by the calibration process and arrays *lower* and *upper* of the same size that define lower and upper limits, so all measured data of the respective characteristic are between these limits. Furthermore, it must be defined how to interpolate between the table values, given the vectors of the independent variables (an $n$-dimensional table is defined by $n$ vectors of independent variables). Often, a linear interpolation is used for tables derived from measurements. In Table 3, examples for this proposed kind of parameterization are given.

**Table 3.** Examples for the proposal to define uncertainty descriptions with generic and tolerance parameterizations of array parameters. $u_1[i], u_2[j], \ldots$ are the vectors of the independent variables, $nominal[i, j, \ldots]$ is the array of the nominal values with respect to $u_1[i], u_2[j], \ldots$. The corresponding array limits are defined with $lower[i, j, \ldots], upper[i, j, \ldots]$ arrays. $ipo$ is the interpolation method to compute the dependent variable $y$ from the independent variables and the nominal array. It is outside the scope of this article to propose a set of standard interpolation methods from the many available algorithms, besides linear interpolation. Default values are given with "$=\ldots$", such as "$relTol = 0$". For example, a 2D uncertainty table might be defined by the vectors of the independent variables $u_1 = [0.1, 0.2, 0.3, 0.4]$ and $u_2 = [10, 20, 30]$, a matrix of size $[4, 3]$ of the *nominal* values $[110\ 120\ 130; 210\ 220\ 230; 310\ 320\ 330; 410\ 420\ 430]$ (e.g., $y = 230$ is the nominal value for $u_1 = 0.2, u_2 = 30$) a matrix of size $[4, 3]$ of the *lower* limits $[11\ 12\ 13; 21\ 22\ 23; 31\ 32\ 33; 41\ 42\ 43]$ and a matrix of size $[4, 3]$ of the *upper* limits $[119\ 129\ 139; 219\ 229\ 239; 319\ 329\ 339; 419\ 429\ 439]$ (e.g., the uncertainty table value of $u_1 = 0.2, u_2 = 30$ must be in the range $[23 \ldots 239]$).

| Uncertainty Name | Parameters |
|---|---|
| *Interval* | $u_1[i], u_2[j], \ldots, nominal[i, j, \ldots], lower[i, j, \ldots], upper[i, j, \ldots], ipo = \ldots$ |
| *Uniform* | $u_1[i], u_2[j], \ldots, nominal[i, j, \ldots], lower[i, j, \ldots], upper[i, j, \ldots], ipo = \ldots$ |
| *TruncatedNormal* | $u_1[i], u_2[j], \ldots, nominal[i, j, \ldots], lower[i, j, \ldots], upper[i, j, \ldots],$ $stdDev, ipo = "linear"$ |
| *IntervalTolerance* | $u_1[i], u_2[j], \ldots, nominal[i, j, \ldots], relTol = 0, absTol = 0, ipo = "linear"$ |
| *UniformTolerance* | $u_1[i], u_2[j], \ldots, nominal[i, j, \ldots], relTol = 0, absTol = 0, ipo = "linear"$ |
| *TruncatedNormalTolerance* | $u_1[i], u_2[j], \ldots, nominal[i, j, \ldots], relTol = 0, absTol = 0,$ $stdDevFactor = 3, ipo = "linear"$ |

For simplicity, it might often be sufficient to define statistical properties by a scalar that holds for all elements, instead of providing statistical properties individually for every array element. For this reason, Table 3 also proposes parameterizations with *relTol* and *absTol* that hold for every array element. For example, the elements of an aerodynamic coefficient table might have an uncertainty of 50% based on previous experience, or the reference results should be reproduced within a relative tolerance of 0.1%.

The question arises how a parameter array can be used to compute the uncertainty distributions of output variables. For scalar parameters, the standard Monte Carlo simulation method is based on the approach to draw sufficient numbers of random numbers according to the respective distributions and for each set of selected parameter values perform a simulation. Such a procedure might not be directly applicable to parameter arrays, because randomly selected elements of an array might give a table that does not

represent physics (for example, if the table output is a monotonically increasing function of the input, then this property might be lost if table elements are randomly selected).

For simplicity, the following (new) method is proposed for 1D tables. A generalization to multidimensional tables is straightforward. A 1D table defines a function $y = f(u)$ that computes the output $y$ from the input $u$ by interpolation in a table. The table is defined by a grid vector $u[:]$, and an output vector $y[:]$, so that $y[i]$ is the output value for $u[i]$. The output $y[i]$ is computed from the nominal value $nominal[i]$, lower and upper limits $lower[i], upper[i]$ by a convex combination defined with uncertain parameter $\lambda$, so that $\lambda = -1$ results in $y[i] = lower[i]$, $\lambda = 0$ results in $y[i] = nominal[i]$ and $\lambda = 1$ results in $y[i] = upper[i]$:

$$y[i] = \begin{cases} \lambda \cdot upper[i] + (1 - \lambda) \cdot nominal[i] & \text{if} \quad \lambda \geq 0, \\ (1 + \lambda) \cdot nominal[i] - \lambda \cdot lower[i] & \text{else.} \end{cases} \tag{8}$$

During the Monte Carlo simulation, random values are drawn for $\lambda$ and new values of the output vector $y[i]$ are computed for every such selection and used in the corresponding simulation. It might be necessary to adapt the lower/upper arrays so that monotonicity requirements are fulfilled by $y[i]$ (e.g., $y[i] \geq y[i-1]$). An example is given in Figure 4, where this table interpolation method is applied to the stiffness of a gearbox described by a rotational spring constant $c$ as a function of the relative angle $\varphi_{rel}$. $y_{\lambda=0.7}[i]$ is the output $y[i]$ for $\lambda = 0.7$.

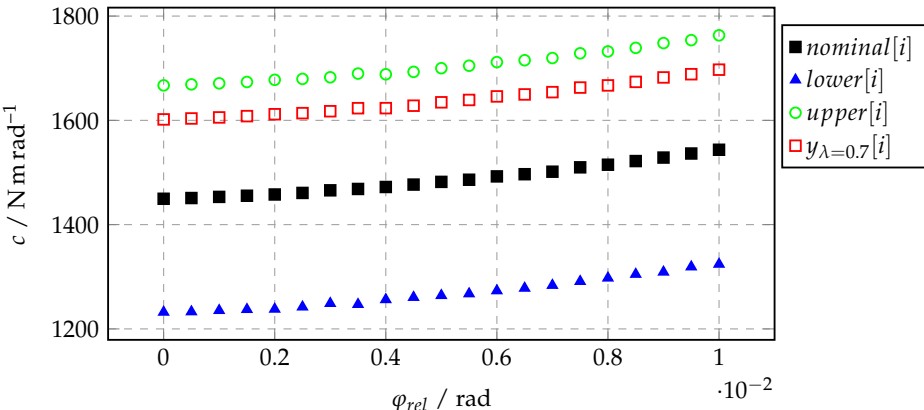

**Figure 4.** Exemplary data of an uncertainty table. Values of $y_{\lambda=0.7}[i]$ are calculated by Equation (8) for $\lambda = 0.7$.

The advantage of this method is that an uncertainty table can be managed in the model just like a standard table, that is, the rules for data interpolation or for extrapolation outside of the $u$ limits apply seamlessly. Moreover, epistemic or aleatory uncertainty definitions can be used for the calculation of $\lambda$. The additional computational effort to compute $y_{\lambda=0.7}[i]$ once per simulation is just marginal.

### 3.3. Model Calibration

Note, this section is a slightly improved version of Section 3.4 of [9]. Once a model is implemented, the modeling process goes on with the calibration, validation and verification, to ensure that the model appropriately fulfills the requirements it was created for. Since the scope of the model calibration, validation and verification is very large, we can only give a short overview in the context of a typical use case for a multiphysical Modelica model. For a broader overview on the topic, a recent paper [11] goes into more details and gives many additional references.

Multiphysical Modelica models are typically used either to represent a real physical system or to reproduce the behavior of such a system as a part of a model-based feedforward or feedback control system. The direct generation of inverse models from Modelica models

is an especially powerful feature which can also be used during the calibration process, see [31,32]. Figure 5 shows an overview of the model calibration process. The different steps are described below.

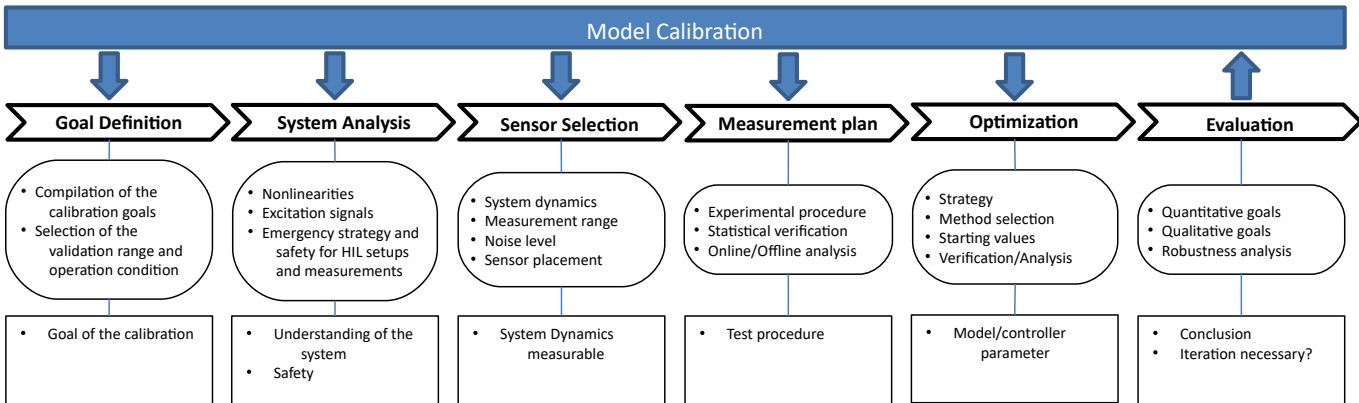

**Figure 5.** Overview for the Modelica model calibration process using measurement data.

### 3.3.1. Goal Definition

The goal of the calibration process is to parameterize models with the help of measurements with regard to defined goals and criteria. For models, this generally means that they should map the behavior of the real-world system as precisely as possible.

The calibration of the parameters of these models is important in order to achieve a good representation of the real system or to have a good controller performance, in the case of model-based control.

As previously discussed in Section 3.1, for many physical systems the knowledge of the individual involved parameters can be quite different. For example, for the model of a robot, the mechanical parameters such as link lengths or masses could be known very precisely from data sheets or CAD data, whereas the friction or damping in connecting joints can be highly unknown. Well-known parameters should, thus, be included in the models directly, in order to reduce the number of unknown parameters for the calibration process. Additionally, the source of these parameters should be documented as described in Section 3.

### 3.3.2. System Analysis

There are numerous approaches for the identification process in the literature. A distinction is made between methods for verification and identification in the frequency domain and in the time domain. Methods in the frequency domain are useful for systems that have (approximately) linear behavior, or for individual operating points of a system for which this assumption applies. Methods in the time domain are also suitable for nonlinear systems, but are usually associated with greater effort (e.g., with regard to the required computing time and experiment effort/duration).

Multiphysics Modelica models are usually used for the modeling of complex nonlinear systems. However, the generation of linear systems from Modelica models is also possible using numerical linearization, which is supported by many Modelica tools. Nonetheless, the focus is on nonlinear models in the following.

In a second step of the calibration, the physical system has to be analyzed after the goal of the calibration has been defined.

Normally, models of a physical system do not contain every detail or the entire operating range of the system. It must, therefore, be investigated to what extent an undesired or nonmodeled behavior can be isolated. For the verification of model-based controllers on (HIL) test benches, further considerations must be made, such as taking precautions to ensure the safe operation of the test bench in case of a controller failure (e.g., unstable behavior or violation of manipulated variable restrictions). In addition, suitable

emergency strategies (e.g., emergency stop switch, mechanical emergency braking) must be implemented. If the controlled system is unstable without a controller, a robust parallel controller can also be helpful, which can be activated in the event of a fault and bypasses the controller to be verified.

### 3.3.3. Sensor Selection

Suitable sensors have to be selected based on the calibration objectives. It is important to consider the dynamics and measuring ranges of the used sensors. If no single sensor can capture the entire relevant dynamics of the system, it must be examined whether a suitable result can be achieved by merging several measurements and/or sensors. Sensors generally have measurement noise and offsets, that must be considered during measurements and appropriately compensated/calibrated. This can also be done during a preprocessing step. After an analysis of the system, the sensor placement must be selected in such a way that the dynamics of the system can be reproduced. This is especially important for elastic mechanical systems (e.g., eigenmode shapes). In case of doubt, different placements should be examined (in the case of elastic systems, the measurements can be influenced by vibration modes, for example).

### 3.3.4. Measurement Plan

After a suitable sensor selection and placement, a measurement plan should be documented, for which the statistical nature of measurements must also be considered. Critical measurements (e.g., measurements of parameters with a large impact on the system dynamics) must always be carried out several times and, if there is a broader spread of the measurement results, they must also be processed appropriately. Particularly in the case of large deviations, the sensor selection and placement must be critically examined again.

For the verification and tuning of controller parameters directly on the test bench, online methods are available, in which the evaluation takes place directly after or already during the measurements on the test bench. HIL setups are suitable for this, in which the Modelica based model controller parameters can be changed with minimal effort.

Alternatively, offline methods can be used by first identifying suitable Modelica models of the controlled system in order to be able to design appropriate controllers with the help of simulations. For the identification of Modelica models and model-based observer systems of the physical system, an "offline" method can always be used, because the change of the Modelica models' parameters does not change the measurement data, which is, on the contrary, the case if the model is part of a feedback system. This enables more elaborate methods to be used. Since Modelica models typically also do not represent the physical system up to very high frequency ranges, noisy measurement data should be low-pass filtered before the calibration process if possible, using forward–backward filtering to avoid a phase shift in the data.

### 3.3.5. Optimization

Suitable identification strategies for Modelica models are optimization-based methods. Therefore, mathematical criteria need to be defined which shall be minimized using a suitable optimization algorithm by varying the parameters of the model. Appropriate start parameters have to be selected for this purpose. An important difference with classical optimization is the statistical nature of the measurement results, which must be appropriately considered for the criteria specification. In the case of HIL optimization, optimization methods with a small number of steps (function evaluations) are typically used, since HIL experiments are usually significantly more time-consuming compared to pure numerical evaluations. This means that local optimization methods are to be preferred (e.g., gradient-based methods or surrogate optimization techniques).

For offline methods, nearly all optimization algorithms can be used. Due to the (possible) noise in the measurement data, convergence is better for gradient-free algorithms. For Modelica models, a wide range of optimization tools are available. There are methods

for optimizing the Modelica model directly within Dymola (https://www.3ds.com/products-services/catia/products/dymola/ (accessed on 30 July 2022)) using, e.g., the DLR *Optimization* library [33], as well as many other external tools. The latter use the Modelica model directly as an executable or exported as an FMU within a chosen environment, such as Python (https://www.python.org/ (accessed on 30 July 2022)) or MATLAB (https://www.mathworks.com/products/matlab.html (accessed on 30 July 2022)), see, e.g., [34].

### 3.3.6. Evaluation

After the Modelica model of the system has been verified, the obtained results must be assessed quantitatively and qualitatively with regard to the selected objectives. To ensure robustness and to avoid overfitting, additional measurements should be used for this step, which were not used within the optimization process of the parameters. If not all goals could be met during the calibration process, the process must be carried out again iteratively after an analysis of the results and, if necessary, a new modified model or controller structure must be used.

As a final result, the obtained set of model parameters should be documented, together with the model and the original measurement data, as well as a detailed description of the overall process. For Modelica models, such a documentation can be done directly within the model, see Section 4.

### 3.3.7. Calibration at the System Level

A technical system typically consists of multiple single components, often delivered by specialized component manufacturers. Analogous to the real-world system, a simulation model for a complete system also consists of different component models, developed by different organizational units. This has an impact on the calibration process described above, because a system model consisting of calibrated component models is not necessarily calibrated. The possible reasons are:

- The behavior of the system leads to a condition the component was not calibrated for.
- Realistic connections between the components need to be modeled. Imagine a contact between two components including friction. The coefficient of friction can only be determined at the system level.
- The influence of uncertainties in the parametrization—negligible for the single component—can be strongly increased by the interdependence with other uncertainties in the system.

As a consequence, a separate calibration of the system is usually necessary.

The fact that the different components and the system are modeled by different developers and the reference data (e.g., measurements) for the calibrations are provided by other persons again leads to various *credibility gates* on the way to a calibrated system model. Each time models, measurements, etc., are handed over to another developer, we call it a credibility gate, because each developer has to query how credible the provided data are. The first gate is between the measurement provider and the component model developer, a second one between the different component model developers and the system model developers and a third one between providers of measurements at the system level and the system model developers. Finally, there is a fourth gate between the system model developers and the user of the system model or its results in the form of a simulation report.

This process, including the gates, are schematically shown in Figure 6. The number of credibility gates may be even higher, because additional independent reference data are necessary for the validation of the calibrated components and of the overall system model.

At each of these gates, the transferred data must fulfill a level of credibility agreed in advance between the transfer partners.

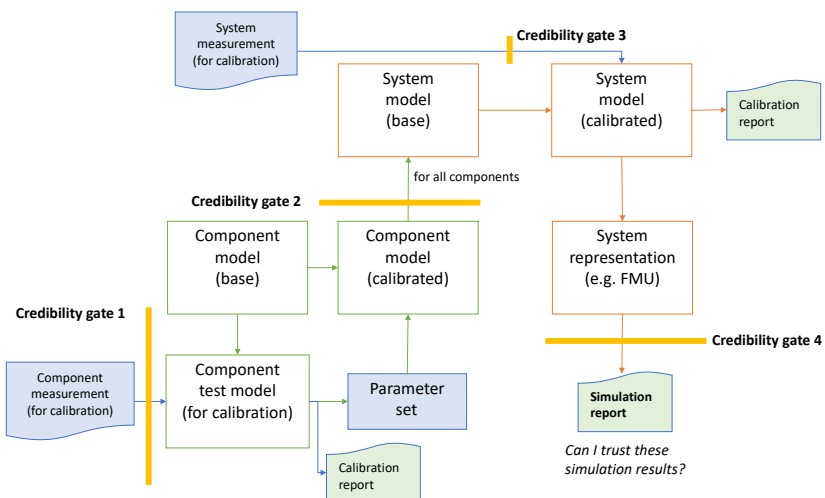

**Figure 6.** Schematic calibration process of a system model.

## 4. Experimental Parameter Credibility in Modelica

The *parameter credibility* proposal of Section 3 was implemented in the new, open-source Modelica library *Credibility* and demonstrated in a use case of a rotational drive train, see Sections 4.2–4.4. Alternative approaches are discussed in Section 4.1.

### 4.1. Potential Approaches

To take full advantage of the elaborated parameter's credibility, the parameter handling of Modelica shall be improved. Especially, a better support of the formal definition and a standardized solution is needed that is supported by all Modelica tools. In particular, the following features would be useful:

- Defining the domain of validity of the model, preferably with a language element and not just in the documentation.
- Define parameter tolerances and/or uncertain distributions (such as normal or uniform distribution).
- Introduce an orthogonal concept to parameter propagation by mapping parameter values, their tolerances and distributions into a model, so that model structure/equations and model data can be more easily separated. This could be done by the *merge* concept proposed in [26].

There are various ways to implement these features:

- In principle, extensions could be introduced via *Custom Annotations* [35], that is, adding additional annotation statements to parameters. Unfortunately, custom annotations are not yet standardized. Furthermore, Modelica tools typically do not provide a graphical user interface for annotations and, therefore, large-scale usage of custom annotations is currently not practical.
- Using an extended variable type inherited from the predefined numerical type "Real" and enhanced by user-defined attributes, such as, e.g., uncertainty. The drawback is a limited reutilization of attributes for multiple parameters and a necessity to handle this within the Modelica language specification. The advantage is that Modelica tools usually have a graphical user interface for variable types.
- The information could be saved outside of a Modelica model, for example, by using SSP elements. The drawback is an increased effort to consistently maintain the model and this extra information.
- A Modelica record collecting credibility attributes of a parameter. The advantage is that a graphical user interface is available, and all data can be defined and stored in a reasonable way. Furthermore, when an FMU is generated, all the credibility information is automatically available in the FMU and accessible from the user of the FMU. The

drawback is that the relationship of the additional data to a parameter is not formally expressed in the language, but this relationship is only defined by "convention".

In the next section, the last proposal is used because it does not require extensions of Modelica tools and a graphical user interface is available.

### 4.2. Modelica Library Credibility

The open-source Modelica library *Credibility* will be made available at (https://github.com/DLR-SR/ (accessed on 30 July 2022)) under the BSD3 open source license. It was newly developed as an initial attempt to provide credible information to Scalar and Table1D parameters, together with a simple model of a controlled drive train to demonstrate how a credible Modelica model can be defined and used. A screenshot of the Credibility library is shown in Figure 7.

Record *Credibility.Scalar*, shown in the upper right part of the figure, contains (i) the real value of the scalar parameter as well as its (ii) traceability, (iii) uncertainty and (iv) calibration information according to Section 3. The simple, nonlinear example model `Credibility.-Examples.SimpleControlledDriveNonlinear.PartialDrive` of a *drive train* is shown in the bottom part of the figure. It consists of the rotational inertia of a motor `inertiaMotor`, an ideal gear `idealGear`, a parallel spring/damper combination `spring/damper` that models the stiffness and damping of the gearbox, the inertia of the load `inertiaLoad` and a driving torque input signal `torqueMotor.tau` as input to this system.

Furthermore, in the upper right part, `data` is a replaceable instance of record `PartialData`. It contains instances of `Credibility.Scalar` for parameters Ja, Jm, ratio, `damping` and `Credibility.Table1D` for parameter `stiffness`, but without concrete values yet. The essential parameters of the drive train reference these records via modifiers such as `inertiaLoad.J = data.Ja.value`. Here, the convention is used that the credibility information defined in `data.Ja` is the credibility information for `inertiaLoad.J` due to the `data.Ja.value` modifier.

The replaceable `data` instance separates the parameterization from the rest of the model. Model `PartialDrive` is valid for a *large range of applications*. When extending from it and redeclaring `data`, as for example done in Figure 8, the model is specialized to *one specific application* (in Figure 8 to the concrete values and concrete credibility information defined in record `DataVariant001`).

In Figure 8, a simple example model of a *controlled drive train* is shown that can be simulated and that allows first experiments.

This model extends from `PartialDrive`, adds a reference, controller, load torque, and redeclares `data` to be an instance of record `DataVariant001`. In the bottom part of the figure, the table matrix is shown defining the nominal spring torque, as well as the upper and lower limits for the spring torque as a function of the relative angle between gearbox and load inertia. The actual table used during the simulation is computed from this table matrix with Equation (8). This model can, for example, be used in Monte Carlo simulations.

Via various parameter menus inside `data`, the following information is provided for the parameters of this model:

- The inertia of the load, as well as the ratio of the gearbox, are assumed to be known from data sheets.
- The motor inertia is computed from CAD data.
- The damping of the gearbox is unknown and only a range for the damping parameter interval is known. The damping parameter is determined by the calibration process described below.
- The nonlinear stiffness of the gearbox is known within a lower and upper bound in the form of a table matrix. It is determined by the calibration process described below.
- The parameters of the feedback controller were determined in a predesign phase not described here. Below, it is verified that the controller performance (error) and the control effort remain within an acceptable 20% range with respect to the nominal design using nominal values for all parameters.

The uncertainties of the parameters were described by truncated normal distributions with the exception of the stiffness, which was described by a uniform distribution. For a real-world system, the available information is usually more limited and the complexity to determine suitable parameter values and parameter uncertainties is much higher.

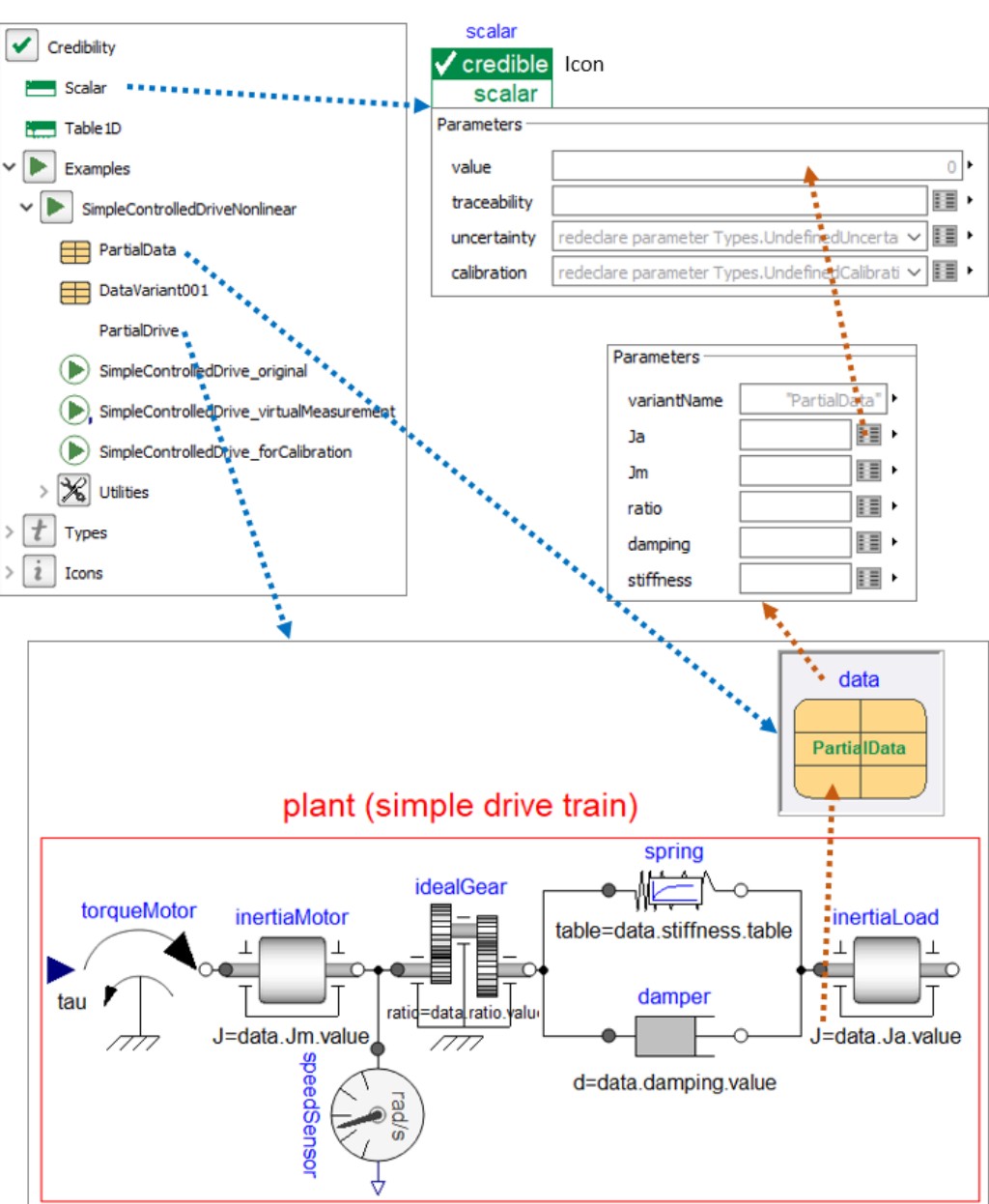

**Figure 7.** A screenshot of Modelica library *Credibility*. In the upper left part, the library is shown and in the lower part, the simple, nonlinear example model `Credibility.Examples.SimpleControlled-DriveNonlinear.PartialDrive` of a drive train. In the upper right part, the parameter menus of instance `data` are shown, which are present in `PartialDrive`. This instance does not yet contain data values for the parameters. Elements `Ja`, `Jm`, `ratio` and `damping` are instances of record `Credibility.Scalar` (shown in the upper right part). Element `stiffness` is an instance of the record `Credibility.Table1D` that is shown in Figure 8.

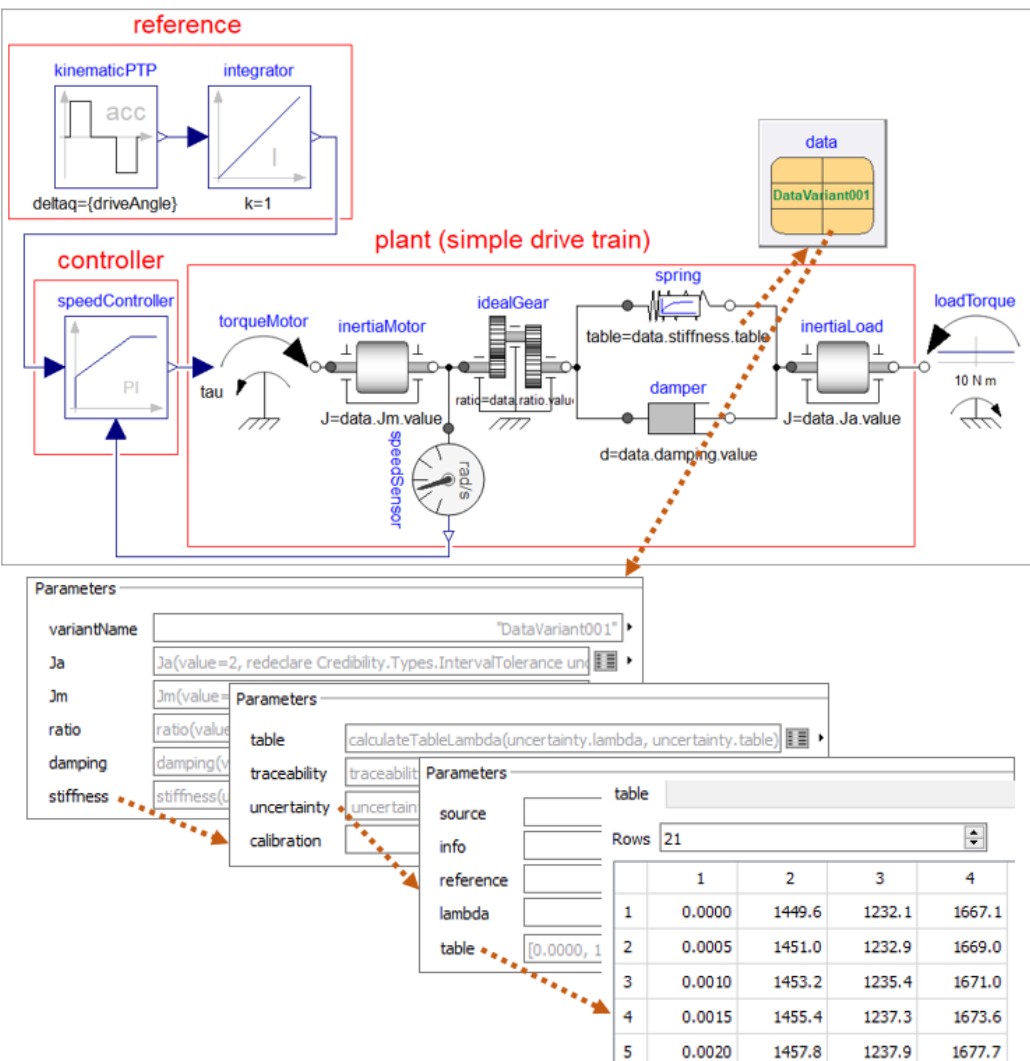

**Figure 8.** A screenshot of component `Credibility.Examples.SimpleControlledDriveNonlinear.-` `SimpleControlledDrive_original`, which is a model that can be simulated, consisting of the drive train together with a reference, controller, data values and a load torque. Additionally, the parameter menus are shown, which define the uncertainty table of the spring stiffness. The first column of the table is the relative angle, the second column is the nominal spring torque, the third and the fourth column are the lower and the upper limits of the spring torque.

### 4.3. Model Calibration

In this section, the calibration of the drive train model is sketched. To keep the example simple, the measurements used for the calibration of the *damping* parameter were generated artificially in a preprocessing step with model `Credibility.Examples.-` `SimpleControlledDriveNonlinear.SimpleControlledDrive_virtualMeasurement` by using the nominal parameter values and an assumed damping parameter value of 100 N m s/rad. To have a sufficient excitation of the damping parameter, a sine sweep with constant amplitude and increasing frequency was used as input for the reference value of the feedback controller. Noise was added to the sensors and the simulated noisy sensor data were stored in a table, which is used below as the virtual measurement for the calibration.

The calibration was performed by tuning the damping parameter with DLR's *Optimization* library [33] so that the integral over the L2-norm of the differences between simulated and measured data of the motor and load angular velocities became as small as possible. The setup of this optimization, as well as the model to be optimized, is shown in Figure 9.

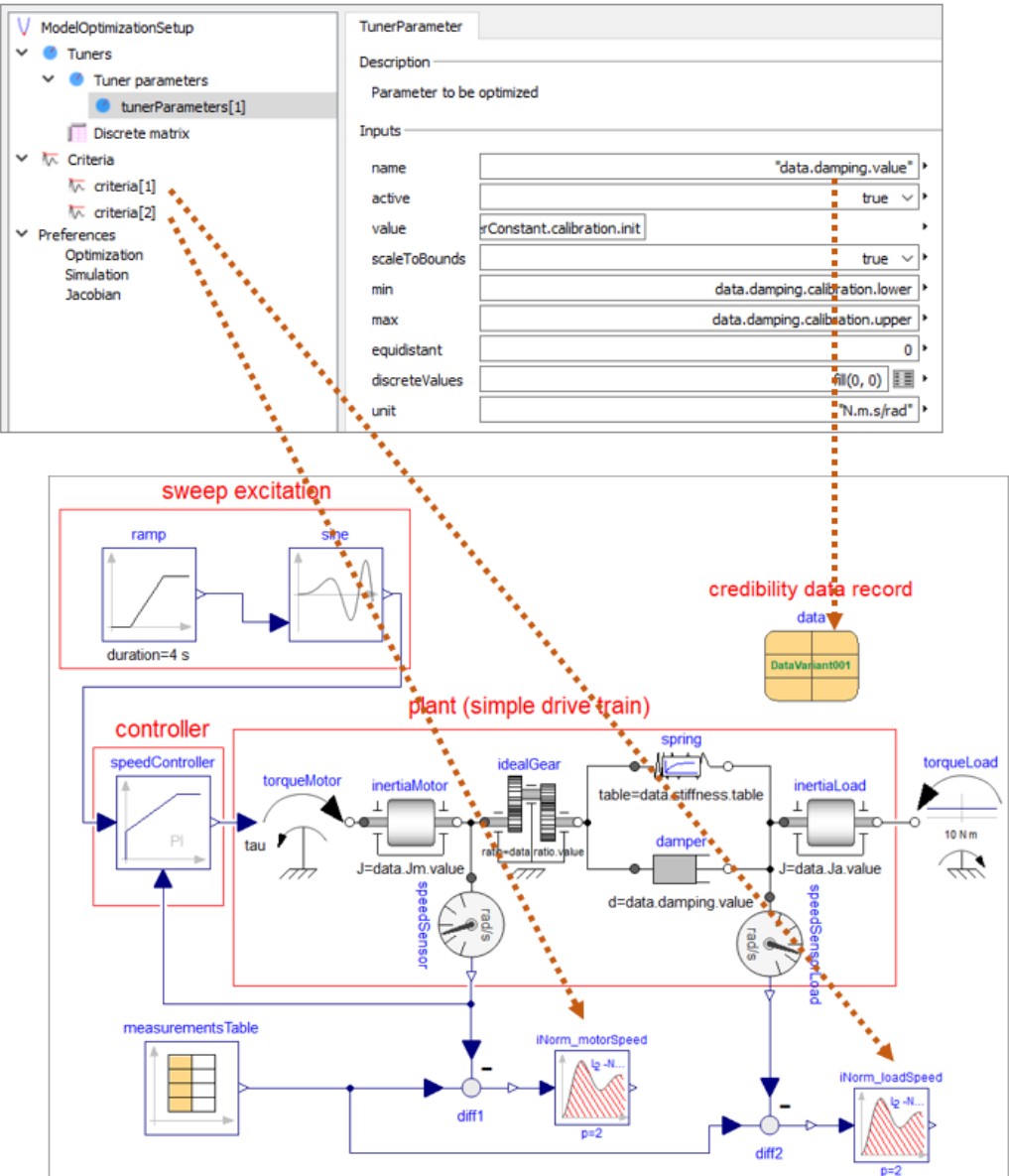

**Figure 9.** The bottom of the figure shows the *model used for calibration*. It consists of a sweep excitation block that generates a sine sweep with constant amplitude and increasing frequency as reference value for the feedback controller. The controller commands a motor torque for the drive train. Two sensors measure the angular velocities of the motor and load inertias. Measurement data are provided as Table `measurementsTable` and the criteria for the optimization-based calibration of the damping parameter is calculated by the integral over the L2-norm of the difference between simulated and measured data. The upper part of the figure shows a screenshot of the user interface for the *calibration setup* with DLR's Optimization Library. The initial value of the damping parameter d, as well as its lower and upper bounds, is directly set by using the information stored in the `data` record of the drive train.

The calibration optimization setup consisted of the nominal model with a starting value for the damping parameter of $300 \, \mathrm{N \, m \, s/rad}$ and an assumed possible range for the parameter of $[1, 1000] \, \mathrm{N \, m \, s/rad}$. The range and starting values for the optimization, as well as the name of the optimization setup model, were stored directly in the credibility record `data` within the model and this information was accessed directly from the optimization setup (see attributes of `damping` in Appendix B, Listing A4). For the optimization, a gradient-free pattern search algorithm [36] was used which is able to cope with the noisy

measurement data. Using this simplified setup, the optimizer returned an optimal value of $99.784\,\mathrm{N\,m\,s/rad}$ for the damping after 25 iterations. For this value an uncertainty of 20% was assumed for the following analysis. The resulting optimal parameter value was then stored (manually) in `data.damping.value` together with its assumed uncertainty.

*4.4. Performance Assessment by Monte Carlo Simulation*

The procedure described above corresponds to the optimization step from Figure 5 and it was then followed by the final evaluation step, which is represented for the simplified example by the following analysis.

To verify the closed-loop performance of the controller, a Monte Carlo simulation was carried out using the MATLAB based DLR multiobjective optimization software environment for control system design *MOPS* [37]. Since a Monte Carlo analysis is widely used, there are many different alternative implementations available, see for example [38,39]. For models with few parameters, a parameter grid search would also be possible. However, this approach does not scale with a higher number of parameters $n_p$ and (discrete) parameter steps $n_s$ because $O\left(n_s^{n_p}\right)$ simulations are needed. As an alternative, an antioptimization approach could also be used [40,41] (i.e., find parameter values so that the system behaves as bad as possible) which can be helpful for complex models to reduce the number of needed simulations.

For the analysis, an executable simulation model `dymosim.exe` was generated from Dymola that was called from MOPS to perform simulations with different values of model parameters. An alternative would be to generate an FMU for a cosimulation that is called from MOPS.

The parameter values and their uncertainties were taken directly from the credibility record (Listing A4) which is part of the model. For all parameters, with the exception of the nonlinear stiffness, a Gaussian (normal) distribution was used. MOPS does not support truncated normal distributions. The standard variances of the parameters were defined so that a value was within the defined limits with a probability of 99.7 %. The differences to the truncated normal distributions defined in the model should therefore have been negligible. MOPS does not support epistemic uncertainties either. Interval uncertainties would be mapped to uniform distributions in MOPS. The nonlinear stiffness and its uncertainty were defined by a table, see Figure 8 and Section 4.2. A uniform distribution of uncertainty parameter `lambdaNLspring` was used in the range $[-1, 1]$.

The resulting sampled normal distributions for the damping parameter `d` and the load inertia `Ja` can be seen in Figure 10. Figure 11 shows the normal distribution of the motor inertia `Jm` and the uniform distribution of uncertainty parameter `lambdaNLSpring`, which was used to calculate the nonlinear stiffness characteristics between the limits defined in the table.

The goal of the Monte Carlo analysis was to verify whether the controller performance (error) and effort remained within 20% of the nominal model (with nominal parameters) with a probability smaller than 0.1%. For the Monte Carlo analysis, the uncertainty parameters were used as described in Listing A4. This ensured that the given controller (with fixed parameters) offered acceptable performance for the uncertainty model without further tuning.

For this simplified example, the control error `ctrlError` was defined as the integral over the absolute difference between the controller reference value and the measured motor angular velocity. The control effort `ctrlEffort` was defined as the integral over the absolute value of the actuator torque, as computed by the feedback controller. For the nominal parameter setup, both criteria `ctrlError` and `ctrlEffort` were normalized to one.

A Monte Carlo gridding run with 10,000 simulations and parameter variations was performed using the distributed computation method implemented in MOPS using multiple CPU cores. For this simple model, a single run for a $4\,\mathrm{s}$ long simulation took a computational time of about $0.431\,\mathrm{s}$ on a single core of an Intel Xeon E5-2687W CPU (https://www.intel.com/content/www/us/en/products/details/processors/xeon.html (accessed on 30

July 2022)) running at 3.00 GHz. MOPS was then used to perform a statistical analysis of the two criteria with the exported Modelica model. Hypothesized gamma distributions were fitted on the resulting criteria computations. Using the gamma distributions, a plot could be set up to calculate the risk that the criteria become larger than 20% as compared to a simulation with nominal values with a probability smaller than 0.1%. Figure 12 shows the result of this calculation for the control effort and Figure 13 for the control error (performance).

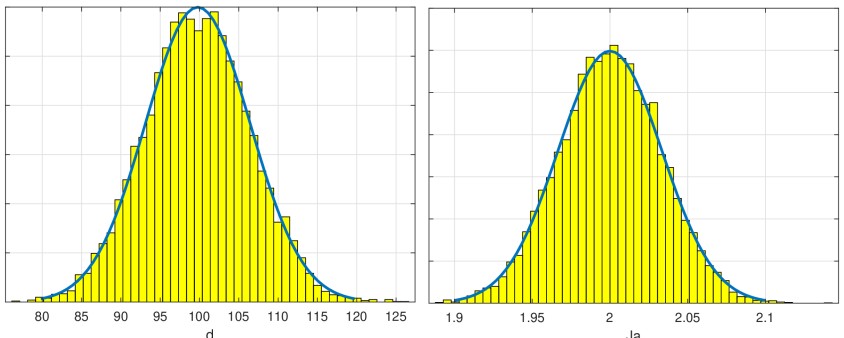

**Figure 10.** Frequency histogram for the normal distribution of parameters `d` (in N m s/rad) and `Ja` (in kg m$^2$) for the Monte Carlo analysis of the example model with 10,000 simulation runs. The plot shows the sampled frequency for each parameter value.

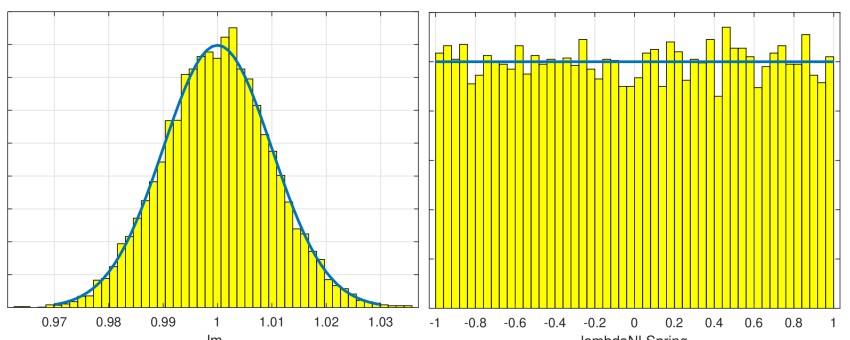

**Figure 11.** Frequency histogram for the normal distribution of parameter `Jm` (in kg m$^2$) and uniform distribution for `lambdaNLSpring` (dimensionless; `lambdaNLSpring` = 0) is the nominal value). The blue line shows the ideal probability density function and the yellow bars the sampled values.

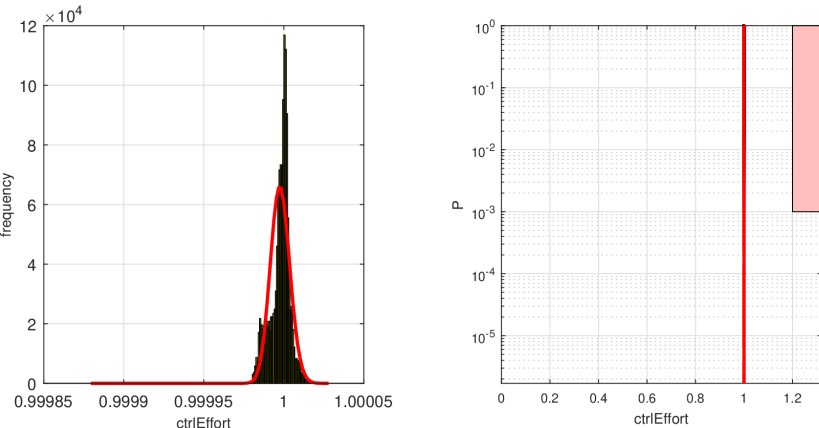

**Figure 12.** The left figure shows the resulting cumulative distribution function (CDF, here a gamma distribution was used) for the model output `ctrlEffort` calculated by MOPS. On the right side, the risk area for the criteria is shown in light red with a black border. The shaded area indicates probability values greater or equal to 0.1%, i.e., the probability calculated from the estimated CDF allows us to conclude the risk is zero.

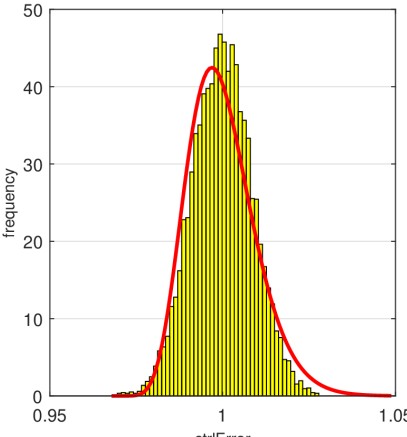 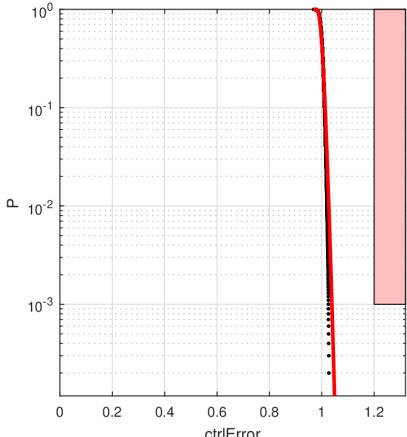

**Figure 13.** The left figure shows the resulting CDF (a gamma distribution was used) for the model output `ctrlError` calculated by MOPS. On the right side, the risk area for the criteria is shown in a light red with a black border. Black dots indicate individual simulation runs and the red line the calculated probability from the estimated CDF.

For the fitted gamma distributions, which show a good match with the simulated data, both requirements can be easily fulfilled. The red block shows the area which would violate the requirements. Both the simulation experiment data (black dots) as well as the fitted probability distribution (red line) show no intersection with the blocks, so the requirements are always fulfilled.

This simple example shows how the Credibility library can be used to store important information on model parameters directly in the model. These data can then be used by Modelica tools, such as the DLR Optimization Library, for a calibration process, or by external tools, such as MOPS, for a verification process. In a similar manner, other external tools (using an FMU or an executable version of the model) and environments such as Python or Julia would also be possible for the calibration and verification process using the information provided by the credibility record.

Having the information directly available in the model helps to keep the information up to date and can allow a simplified (and possibly automated) access to important credibility or calibration data. This is especially helpful when transferring the model to another user.

## 5. Conclusions and Outlook

In this article, concrete proposals were presented to improve the development process of Modelica models to ultimately arrive at Modelica models with documented, standardized quality measures. We found that it seemed useful to include machine-readable metadata into Modelica models, instead of external data formats. This allowed for simplified process steps such as calibration, verification and validation or Monte Carlo simulation.

The credibility information at the model level could be automatically filled by tools, to a large extent. These embedded metadata will help when transferring models across organizational borders, which became a standard use-case due to standardized models in Modelica, FMI and SSP.

The core proposals for parameter credibility were summarized in a new open-source Modelica library *Credibility* in which traceability, uncertainty and calibration information for scalar and array parameters can be defined. The proposals and the application of the new Modelica library in a credible modeling and simulation process were demonstrated with a simple model of a drive train.

The proposals and examples in this article were implemented with standard Modelica language elements, so that current Modelica tools could be utilized. We plan to discuss these proposals in the Modelica Association and with Modelica tool vendors to arrive at better tool support. For example, it should be possible for an external calibration tool to be

able to extract the needed information from a Modelica model and update a Modelica model with the result of the calibration, if desired by the user. Currently, such operations might require manual copying of parameter values, which is not practical for realistic applications. Furthermore, both DLR-SR and LTX plan to evaluate the proposed approach in real-world simulation projects and adapt and improve it if necessary. Finally, the approach needs to be integrated into other developments of the UPSIM project that are currently ongoing.

**Author Contributions:** Conceptualization, M.O., M.R., J.T., L.G. and M.S.; methodology, M.O., M.R. and J.T.; software, M.O., M.R. and J.T.; calibration and performance assessment, M.R.; writing—original draft preparation, M.O., M.R., J.T., L.G. and M.S.; writing—review and editing, M.O., M.R., J.T., L.G. and M.S.; visualization, M.O., M.R., J.T., L.G. and M.S.; supervision, M.O. and L.G.; project administration, M.O. and L.G.; funding acquisition, M.O. and L.G. All authors have read and agreed to the published version of the manuscript.

**Funding:** This work was organized within the European ITEA3 Call6 project UPSIM https://itea3.org/project/upsim.html (accessed on 30 July 2022) — Unleash Potentials in Simulation (number 19006). The work was partially funded by the German Federal Ministry of Education and Research (BMBF, grant numbers 01IS20072H and 01IS20072G).

**Data Availability Statement:** The Modelica library *Credibility*, described in detail in Section 4.2, will be made publicly available at (https://github.com/DLR-SR/ (accessed on 30 July 2022)) under the BSD3 open-source license.

**Acknowledgments:** We would like to thank our colleague Andreas Pfeiffer for their idea of a convex combination to interpolate the uncertainties of arrays as well as for their valuable contribution during discussions and the realization of the model calibration of the simple drive train model. We also highly appreciate their review of this paper and quite a lot of constructive improvement proposals. We would also like to thank Lars Mikelsons for hints on some key papers on uncertainty quantification and Daniel Bouskela who gave us a short explanation of how credible simulations are defined and used for French nuclear power plants (see [18]).

**Conflicts of Interest:** The authors declare no conflict of interest.

## Abbreviations

The following abbreviations are used in this manuscript:

| | |
|---|---|
| CAD | Computer-aided design |
| CDF | Cumulative distribution function |
| CFD | Computational fluid dynamics |
| CPU | Central processing unit |
| DOI | Digital object identifier |
| eFMI | Functional Mock-up Interface for embedded systems |
| FEM | Finite element method |
| FMU/FMI | Functional Mock-Up Unit/Interface |
| HIL | Hardware-in-the-Loop |
| ISBN | International Standard Book Number |
| LFT | Linear fractional transformation |
| LOTAR | LOng Term Archiving and Retrieval |
| MOPS | Multi-Objective Parameter Synthesis |
| MSL | Modelica Standard library |
| OEM | Original equipment manufacturer |
| SSP | System Structure and Parameterization of Components for Virtual System Design |
| SRMD | Simulation Resource Meta Data |

## Appendix A. Modelica Proposal of Model's Metadata

**Listing A1.** Modelica code of user's metadata as currently stored in Dymola's vendor annotation.

```
model MetaDataTest
...
annotation (
...
__Dymola_UserMetaData={
{"Author*", "John Doe"},
{"Organization*", "My Company"},
{"Contact", "john.doe@mycompany.domain"},
{"CreationDate*", "2022-07-04"},
{"ModifiedDate", "2022-07-29"},
{"CreatedInProject", "UPSIM 2022"},
{"License", "BSD 3-clause"},
{"MSLVersion", "4.0.0"},
{"CopiedFromModel", "Modelica.Electrical.Machines.Examples.Transformers.
    Rectifier6pulse"},
{"CopiedFromModelVersion", "4.0.0"},
{"CredibilityLevel", "0"},
{"Checked", "false"},
{"CheckedBy", ""},
{"ModelicaTool", "Dymola"},
{"ModelicaToolVersion", "2023"},
{"ModelicaLanguageVersion", "3.5"}});
end MetaDataTest;
```

## Appendix B. Modelica Implementation of the Credibility Library

Please note, the Modelica library described below was a work in progress at the time of writing this paper and changes could have occurred when the library was made public.

**Listing A2.** Modelica code of credibility records `Scalar` and `Table1D` and their exemplary usage in data set `Data`.

```
package Credibility
record Scalar "Record collecting credibility information for a scalar real
    parameter"
replaceable parameter Real value = 0 "Value of parameter";
parameter Types.Traceability traceability;
replaceable parameter Types.UndefinedUncertainty uncertainty
constrainedby Types.UndefinedUncertainty "Uncertainty";
replaceable parameter Types.UndefinedCalibration calibration
constrainedby Types.UndefinedCalibration "Calibration setup";
end Scalar;

record Table1D "Record collecting credibility information for a real table"
parameter Real table[size(uncertainty,1), 2] =
calculateTableLambda(uncertainty.lambda, uncertainty.table)
"Scaled table";
parameter Types.Traceability traceability "Traceability";
replaceable parameter Types.Interval1D uncertainty
constrainedby Types.Interval1D "Uncertainty"
annotation (choicesAllMatching=true);
parameter String calibration = "" "URI of calibration setup script";
protected
function calculateTableLambda "Calculate scaled table"
input Real lambda;
input Real uncertainty[:,4] "Table uncertainty (x, y, yL, yU)";
output Real table[size(uncertainty,1),2] "Value of table";
algorithm
table[:,1] := uncertainty[:,1];
if lambda >= 0 then
table[:,2] := lambda * uncertainty[:,4]
+ (1 - lambda) * uncertainty[:,2];
```

```
else
table[:,2] := (1 + lambda) * uncertainty[:,2]
- lambda * uncertainty[:,3];
end if;
end calculateTableLambda;
end Table1D;

package Types
record BaseUncertainty1D "Basic uncertainty definitions of 1D tables"
constant UncertaintyKind1D kind "Kind of 1D uncertainty";
parameter SourceType source = SourceType.Unknown "Source of uncertainty";
parameter String info = "" "Information about the source";
parameter String reference = "" "Reference of the source";
parameter Real lambda(min=-1, max=1) = 0
"Convex scaling of table between -1 and 1 (= 0: nominal)";
parameter Real table[:,4] = zeros(1,4)
"Table matrix (columns: 1 = input; 2 = nominal values;
3 = lower limits, 4 = upper limits)";
end

record Interval1D "Interval uncertainty of lambda (epistemic uncertainty)"
extends BaseUncertainty1D(
kind = UncertaintyKind1D.IntervalUncertainty1D);
end Interval1D;
end
end Credibility;

record PartialData "Record of credible parameters of simple control drive"
parameter Credibility.Scalar Jm(
redeclare parameter Modelica.Units.SI.Inertia value = 2,
traceability(
source     = Types.SourceType.Provided),
redeclare Types.IntervalUncertaintyByTolerance uncertainty(
unitValue = "kg.m2",
nominal    = 2,
relTol     = 0.05,
source     = Types.SourceType.Provided),
redeclare Types.Calibration calibration(
unitValue = "kg.m2",
start      = 2,
min        = 1.2,
max        = 3.0,
__ToolA_Setup = "Credibility.Examples.SimpleControlledDrive.
    runSimpleControlledDriveOptSetup"))
"Inertia of motor";
parameter Credibility.Table1D stiffness(
uncertainty = [
0.00, 1.0, 0.0, 2.2;
...
0.10, 1.5, 0.6, 1.8])
"Table for nonlinear spring stiffness in N.m/rad of drive compliance";
...
end PartialData;
```

**Listing A3.** Exemplary usage of credibility record `Table1D` for table-given nonlinear characteristics of a rotational spring. Record `PartialData` used in the `PartialDrive` model is set in Listing A2.

```
model SpringNonLinear "Nonlinear rotational spring"
parameter Real table[:, :]
"Spring stiffness [N.m.rad-1] (col. 2) vs. deflection angle phi [rad] (col.
    1)";
Modelica.Blocks.Tables.CombiTable1Ds stiffness_nonlin(
table = table)
"Force look-up table";
...
end SpringNonLinear;
```

```
model PartialDrive "Not calibrated drive model"
replaceable parameter PartialData data "Data of the drive train";
Modelica.Mechanics.Rotational.Components.Inertia inertiaMotor(
J = data.Jm.value)
"J from datasheet; Init defined by system spec or scenario spec";
SpringNonLinear spring(
table = data.stiffness.table)
"table from datasheet; Init defined by system spec or scenario spec";
...
end PartialDrive;
```

**Listing A4.** Modelica code of credibility record for the simple use case. Record `PartialData` is set in Listing A2.

```
record DataCalibration "Calibration of Data variant 001"
extends PartialData(
variantName = "Calibration of Data variant 001",
Ja(
value = 2,
traceability(
source    = Types.SourceType.Provided,
info      = "Data taken from data sheet",
reference = "modelica://MyProject/Data/Var001/DataSheet.pdf"),
redeclare Types.NormalUncertainty uncertainty(
unitValue = "kg.m2",
mean      = 2,
stdDev    = 0.0333,
source    = Types.SourceType.Provided)),
Jm(
value = 1,
traceability(
source    = Types.SourceType.Computed,
info      = "Data computed form CAD data",
reference = "modelica://MyProject/Data/Var001/CADmodel.stp"),
redeclare Types.NormalUncertainty uncertainty(
unitValue = "kg.m2",
mean      = 1,
stdDev    = 0.01)),
ratio(
value = 2,
traceability(
source    = Types.SourceType.Provided,
info      = "Data taken from data sheet",
reference = "modelica://MyProject/Data/Var001/DataSheet.pdf"),
redeclare Types.IntervalUncertaintyByTolerance uncertainty(
unitValue = "1",
nominal   = 2,
relTol    = 0,
source    = Types.SourceType.Provided)),
damping(
value = 99.78375,
traceability(
source    = Types.SourceType.Calibrated,
info      = "Calibrated from virtual measurements with noise",
reference = "Credibility.Examples.SimpleControlledDriveNonlinear.
    SimpleControlledDrive_virtualMeasurement"),
redeclare Types.NormalUncertainty uncertainty(
unitValue = "N.m.s/rad",
mean      = 99.78375,
stdDev    = 6.6523),
redeclare Types.Calibration calibration(
unitValue = "N.m.s/rad",
start     = 300,
min       = 1,
max       = 1000,
__ToolA_Setup = "CredibilityCalibration.Examples.
    SimpleControlledDriveNonlinear.runSimpleControlledDriveOptNLSetup")),
stiffness(
```

```
traceability(
source    = Types.SourceType.Measured,
info      = "Based on measurements on test bed B in Example project",
reference = "modelica://MyProject/Data/Var001/SpringMeasurement.pdf"),
uncertainty = [
0.0000, 1449.6, 1232.1, 1667.1;
...
0.0100, 1543.6, 1324.1, 1763.1]));
end DataCalibration;
```

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
