# Peer review of "Towards Modelica Models with Credibility Information"

_electronics, doi:10.3390/electronics11172728_

Round 1

Reviewer 1 Report

This manuscript followed the work presented in International Modelica Conference 2021, and proposed the concrete proposals to improve the development process of Modelica models. Overall, the topic of this study is interesting, and the manuscript was well organised. The detailed comments are shown below.

1.       The contribution and innovation of the manuscript should be clarified clearly in abstract and introduction.

2.       Broaden and update the literature review to better connect to the current effort in the field of Modelica models.

3.       A parametric study is necessary for the proposed model.

4.       How is the running time of the proposed method?

5.       More future research should be included in conclusion part.

Author Response

0. Moderate English changes required 

The pdf-file was transformed to Word and the spell and grammar checker of Word was used to improve the manuscript. Additionally, the manuscript was specifically inspected and improved with respect to the English language.

1. The contribution and innovation of the manuscript should be clarified clearly in abstract and introduction.

Done.

2. Broaden and update the literature review to better connect to the current effort in the field of Modelica models.

Done.

3. A parametric study is necessary for the proposed model.

A typical real-world model has many parameters, so that parameteric studies are typically not practical. It is not the intention of the article to provide an overview of the various approaches to cope with this problem. Instead, one often used approach - Monte Carlo simulation - is used as demonstration how the credible model information can be utilized. This has been more clearly described in the manuscript.

4. How is the running time of the proposed method?

Added.

5. More future research should be included in conclusion part.

Done.

Reviewer 2 Report

This paper discusses the Modelica model in documented and standardized quality measures. Credibility is very important for a simulation model, the commonly used verification and validation in simulation modeling are all for the model credibility.  But the term of Modelica is not well known in the field of system modeling and simulation. For example, I cannot find any publication and discussion of the Modelica model in the annual winter simulation conference. It is therefore suggested for the authors to include an introduction and definition of the Modelica and Modelica model, and compare them with the common used methods and simulation software systems in the field of system modeling and simulation.

Author Response

Does the introduction provide sufficient background and include all relevant references? - Must be improved

This paper discusses the Modelica model in documented and standardized quality measures. Credibility is very important for a simulation model, the commonly used verification and validation in simulation modeling are all for the model credibility.  But the term of Modelica is not well known in the field of system modeling and simulation. For example, I cannot find any publication and discussion of the Modelica model in the annual winter simulation conference. It is therefore suggested for the authors to include an introduction and definition of the Modelica and Modelica model, and compare them with the common used methods and simulation software systems in the field of system modeling and simulation.

A short introduction to the Modelica language was added to the introduction, together with references to (a) the Modelica Language Specification, (b) a Modelica overview in the CRC Handbook of Dynamic System Modeling and (c) a comparison of Modelica with other multi-domain modeling languages in the Encyclopedia of Systems and Control. The introduction was further enhanced with an improved description of the contribution and an improved literature review.

Round 2

Reviewer 2 Report

The authors have addressed my questions in the revised paper. I don’t have any further question.